# Blended Learning Reimagined: Teaching and Learning in Challenging Contexts

**Shanti Divaharan *** and **Alexius Chia**

National Institute of Education, Nanyang Technological University, Singapore 637616, Singapore
* Correspondence: shanti.divaharan@nie.edu.sg

**Abstract:** The COVID-19 global pandemic has caused disruptions around the world with devastating consequences socially and economically. Education was not spared. Schools and institutions of higher learning (IHLs) had to grapple with new sets of demands. With most countries forced into lockdown to stem the spread of the virus, some turned to technology-mediated learning to provide some kind continuity for learning to still take place. This concept paper will share some of the key learning points and strategies culled from experiences having to pivot almost overnight to embracing technology and new learning environments, which were sometimes remote or neglected in a milieu and culture that often prided itself in effective physical face-to-face interactions. This article will then draw upon how the Blended Learning approach, undergirded by Connectivism, was implemented in a local IHL. Examples of the different types of blended learning designs that were employed will be described alongside examples on how educators can distinguish between them to engage their learners in both modes.

**Keywords:** blended learning; COVID-19; digital learning environments

## 1. Introduction

When COVID-19 was first detected at the end of 2019, no one expected its voracity, much less its staying power. Since its onset, the world has experienced devastating social and economic disruptions. Education was not spared either. The disruptions caused by the pandemic affected more than 1.7 billion learners, including 99% of students in low- and lower-middle-income countries [1,2]. Education experts argue that time spent away from school could lead to a number of adverse effects on children caused by school closures, a lack of resources for learners to properly do home-based learning, emotional stress and fear, financial hardships on the part of their parents, the learners themselves having to take up employment, a lack of reliable information about the pandemic as well as school matters, and inadequate teacher training to cope with the crisis [3]. All these, Tsolou, Babalis, and Tsoli (2021) [4] opine, increase learners' chances of further exclusion from school and even dropping out and could eventually lead to them being excluded at a societal level. They argue that these conditions—like those outlined by Hallgarten [3]—may negatively impact learners' physical development mental and general health, and access to friends and social networks.

There has been a host COVID-19 related research that has been published recently. And these focused on very varied domains: isolation and the mental health of children and adolescents [5,6], teachers' experiences of stress and coping strategies during COVID related distance teaching [7], limitations of eLearning tools and platforms for access [8], the need to strengthen curriculum and to make it more responsive to the learning needs of the learners in Higher Education institutes [9] and skill levels of teachers and students as well as hardware and software provisions [10] (Eickelmann and Drossel, 2020 as cited in König et. al., 2020). Adedoyin and Soykan (2020) [11] drew attention to the fact that during the pandemic, universities showed a lack of proper planning, design and development of

online instructional programs—a point [3] alluded to as well. Salas-Valdivia and Gutierrez-Aguilar (2021) examined the key factors that allowed successful learning to occur in e-learning during the COVID-19 pandemic [12]. Efthymiou and Zarifis (2021) researched 154 learners of different nationalities in their program and discussed learner difficulties from different parts of the world and emphasized the need for a different set of pedagogies for online learning [13]. Doukanari et al. (2021) investigated the prospects of implementing multidisciplinary and multicultural student teamwork as a form of sustainable learning and how existing and developing technologies were able to support the process [14].

This conceptual paper will first provide the backdrop to Singapore's education system and its push for integrating technology into teaching and learning. It will then discuss how the concept of blended-learning had to be re-looked in order to maintain continued education for our learners due to the various challenges brought about by the COVID-19 pandemic. Two lesson outlines—one pre-COVID-19 and the other conducted during the pandemic—will be presented as exemplars to highlight the changes made to lesson design. The adoption of sound pedagogical design principles adapted from Blended Learning approach and Connectivism will be foregrounded to illustrate the possibilities of ensuring continued learning and engagement during the lockdown. The intention of this paper is not to propose a new theory; rather it seeks, as Cropanzano (2009) so succinctly puts it, to bridge existing theories, in interesting ways, link work across disciplines, provide multi-level insights, and broaden the scope of our thinking [15]. As suggested by Gilson and Goldberg (2015), a good conceptual paper may also build on existing theories by offering propositions regarding previously untested relationships [16]. In the case of this paper, the intention is to explicate the key adaptations made to Blended Learning approach and Connectivism and to encourage educators to broaden their scope of thinking when designing for digital mediated learning.

## 2. Background

While the situation in Singapore was not as acute compared to other parts of the world, to date, out of a population of about 5.8 million people, Singapore has seen close to 1.8 million cases of infections and more than 1500 deaths [17]. Other than a partial lockdown (circuit breaker) for a month from 7 April 2020—where, except for essential services, businesses and workplaces were forced to close and schools pivoted to home-based learning (HBL), the country has had several cycles of opening and tightening where the population was made to adhere to a strict regime of physical distancing, safe management measures, and the wearing of masks. Schools and institutions of higher learning (IHLs) had to resort to creative ways of addressing the partial lockdown and various measures that kept learners out of schools and learning spaces. This situation has forced a re-think of how teaching and learning could still continue despite the sometimes-insurmountable obstacles.

Since 1997, the Singapore government has committed to investing in integrating Information and Communication Technology (ICT) into Singapore schools to transform teaching and learning [18]. The first Master Plan for ICT which was rolled out in 1997, focused on building up the infrastructure and basic ICT skills of teachers and students. The second was introduced in 2003 and the objective was to ensure pervasive use of technology in teaching and learning and to ensure that schools had baseline standards for technology adoption. There was continual investment in building the infrastructure to support mobile technology. This came in the form of investing in wireless infrastructure to support seamless learning environment in the schools and this was achieved under the third Master Plan for ICT in education spanning from 2009 to 2014. By the time the fourth Master Plan for ICT in education was introduced in 2015 and lasting till 2019, schools were well equipped to conduct technology-mediated learning in a seamless manner and teachers and schools were equipped with relevant technology skills sets. While every teacher was provided with a personal device for teaching related purposes, students were already using learning devices such as laptops or tablets issued by the school. By the time the COVID-19 pandemic struck, we were embarking on our Educational Technology Plan. The Educational Technology

Plan consists of a broad vision spanning a decade from 2020–2030. It largely focuses on developing infrastructure, technology tools and associated pedagogy to enable students to take ownership of their learning. While the plan focuses on the provision of a connected learning experience, it also aims to create personalized learning to cater to the individual needs of the students.

Negotiating the various challenges that COVID-19 posed, led to the realization that pockets of inequality still existed across Singapore. Schools soon discovered that many learners did not have access to internet connectivity at home and that their families did not have to finances to provide them with learning devices [19]. To alleviate the situation, the Ministry of Education had to loan out about 12,500 laptops and tablets and a further 1200 devices to enable internet connectivity [20].

## 3. The National Institute of Education, Singapore

The National Institute of Education (NIE), Singapore, is Singapore's sole teacher education institute. It is an autonomous institute of Nanyang Technological University (NTU) with a 70-year history. NIE delivers a suite of initial teacher preparation programs (ITP) that prepares teachers for the Singapore education service. These programs range from a four-year Degree program which offers a BA/BSc (Education) to a 16-month Postgraduate Diploma in Education program to a one to two-year Diploma program. Besides ITP, NIE also offers various graduate programs at the masters and doctoral levels as well as in-service programs which caters to a wide spectrum in the education community.

*Our Learning Environment*

At NIE, we have relevant wireless technology infrastructure in place. While our faculty have laptops provided to support the teaching and learning, the students bring their own devices. Digital Learning Environments (DLE) are spaces that bring together the teacher, students, and technology tools with the purpose of creating a technology-mediated environment to support effective teaching and learning [21]. Within the teaching and learning environment in our IHL, DLE has been implemented pervasively. With the partial lockdown and ensuing periods where faculty and students had to physically stay away from campus, it necessitated a move away from the technology-mediated learning environment within the physical campus and to adopt blended learning—defined as combining face-to-face instruction with an online mode of learning [22–24]—as an approach to ensure that learning continued to take place. One of the challenges brought about by the pandemic was that physical learning spaces became non-existent overnight. As such, virtual learning spaces had to be used as alternatives to physical spaces—as advocated by the blended learning approach. A virtual learning space or a virtual classroom is an online environment where teachers are able to connect with their learners in situations where a physical learning space was not available [18,25,26]. Virtual learning spaces are useful platforms for synchronous learning to take place where teachers and their learners interact in real-time online. This is in contrast to an asynchronous mode of learning, where the learners can access a pre-prepared set of resources that are uploaded on a digital platform at their own convenience without the need to be online with their teachers at the same time [18,25,27].

## 4. Types of Blended Designs

In Table 1, Graham (2006) outlines four types of blended learning designs. They are (i) activity-level blending where learning takes place in a face-to-face environment and with elements of technology-mediation to support learning activities. (ii) course-level blending when there is a clear distinction made between learning in a face-to-face environment and online learning or learning in a virtual space. (iii) program-level blending usually occurs in IHLs where learners choose to attend a mixture of both face-to-face and online courses. (iv) institutional-level blending where the institution offers classes at the

beginning and at the end of the courses and in between, the students learn the content online asynchronously [22].

**Table 1.** Graham's (2006) Blended Learning Designs.

| Blended Learning Designs | | | |
|---|---|---|---|
| **Activity-Level Blending** | **Course-Level Blending** | **Program-Level Blending** | **Institutional-Level Blending** |
| Learning takes place in a face-to-face environment and with elements of technology-mediated learning | Clear distinction is made between learning in a face-to-face environment and technology-mediated learning (online learning) | Usually occurs in IHLs, where students choose to attend a mixture of both face-to-face and online courses to complete their education | Institutions offer classes at the beginning and at the end of courses and in between, students learn the content online |

Prior to the COVID-19 pandemic, technology-mediated learning for in-person lessons and blended learning approaches were the most common modes used at NIE. However, with the campus physically closed during the partial lock-down and ensuing months of safe management measures where lessons had to take place offsite, these modes were no longer viable to sure effective delivery of our programs. To this end, we had to adapt the blended learning approach from a blend of physical and online asynchronous learning space to a blend of virtual learning space and online asynchronous learning space. The next section will explicate the design considerations adopted in implementing activity-level blending and course-level blending during these challenging times. Given the need to move into a virtual learning space, it was prudent to note the following advantages of teaching in a synchronous online learning environment (adapted from [28]).

1. Being online together and learning together provides motivation for students.
2. While it is not the same as a physical classroom setting, by interacting online with one another, group cohesion among the students and a sense of belonging can be fostered.
3. Being in a synchronous learning environment provides opportunities for immediate feedback, support and decision-making in group activities.
4. Helping to pace the lesson activities encourage the student to be disciplined in learning and to prioritize their studies.

With online virtual teaching and learning occupying a much larger space in the ways universities deliver the curriculum, there were a number of considerations that had to be taken into account. For one, how lesson designs would engage the students, ample opportunities for interactivity (not only with each other but also with content) and making sure that there was minimal screen fatigue. Another consideration was, in cases where was a choice, how one would decide if a lesson were an online (asynchronous) one or a virtual (synchronous) one. The next section will outline the different types of blended learning designs that were employed at NIE and provide some examples on how educators can distinguish between them so as to engage their learners in both modes.

*4.1. Example of a Lesson (Conducted Pre-COVID-19)*

Table 2 is a description of a lesson from a doctoral program. The lesson required graduate students to unpack the key characteristics of a qualitative methodology, Grounded Theory, and to be prepared for a discussion in class. They were then required to apply their understanding of the characteristics of the methodology and to critique a journal article.

*4.2. Example of a Lesson (Conducted during the Pandemic)*

The lesson described in Table 2 was subsequently modified in order to adapt to the changed learning environment (see Table 3). Having to attend lessons from home (HBL), many of the in-class activities which was described in the previous section was no longer possible. It is evident from the modified lesson design that HBL has increased the dependence on technology-mediated learning. The key question in working on this new

design were: How will the lack of physical learning space and physical interaction impact the design of Blended Learning Approach? Given the fact that the use of technology would be prevalent to enable effective learning to take place, the design of blended learning had to be re-examined. What then were the key design considerations that needed to be factored into refining the lesson to: (i) reduce online fatigue; (ii) to ensure the learners had ample opportunities to interact and participate as they would in a face-to-face environment; and (iii) tutors had sufficient opportunities to monitor the learners' progress as they would in a face-to-face setting?

**Table 2.** Lesson Outline: Understanding key characteristics of 'Grounded Theory' as a qualitative methodology.

| Lesson Segment | Description | Resources |
|---|---|---|
| Pre-class learning (online, asynchronous) | Tutor assigns relevant methodology readings on Grounded Theory (GT). Additional readings in the form of journal articles are also assigned. Short video clips are assigned to provide a quick introduction to GT. | Pre-class learning (online, asynchronous, use of Learning Management System) |
| Blended Learning Approach: Course level Blending—Graduate students are given opportunities to immerse with the resources at their own time | | |
| During class | Tutor recaps clarifies questions raised from the readings and viewing of the video. Tutor highlights and reiterates key characteristics of GT. In their working groups, graduate students work on applying what they have learnt to an article which they are required to critique. Tutor walks around and helps to clarify doubts and questions which might arise. Tutor helps to facilitate the groups so that they are focusing on the salient points of GT as a research methodology. It is possible to integrate Activity-level blending within this lesson design. | Slides for content input, summarizing salient features of the methodology, flip chart paper for graduate students to highlight evidence from the article that match the research design. They are to prepare to share their discussion explicating the key characteristics of GT as a methodology adopted in the article. |
| Blended Learning Approach: Activity-level Blended Learning—Graduate students have the opportunity to access the online resources, while working in groups, to reference the resources in the event that they have doubts about what the key characteristics of GT as a research design. | | |
| During class | Group presentation of their critique of the article. Feedback received from tutor and peers about the points that they have raised. | Flip chart, group presentation and face-to-face interaction, peer learning |
| Post-class | Focus is on the application of what the Graduate students have learnt during tutorial, and they have the opportunity to design for a possible area of research applying GT design principles. | Graduate students work on their own, at their pace. They can access the materials and resources posted by the tutor for the pre-tutorial activity as well. |

Design Considerations Undergirded by Connectivism

Connectivism is "social learning that is networked" [29] (p. 6). It is based on the principle that all learning starts with a connection [30]. These connections occur on a neural, conceptual, and social levels [31]. The proliferation of technological connectivity in the 21st century has enabled communities to collaborate in multiple ways on a wide range of topics and domains resulting in a collective network which connects local and international communities [29]. The fundamental shift in how we see learning communities through the creation of these social networks have prompted educators to embrace this new mode of knowledge propagation [29,30]. There are a few principles of connectivism which Bell lists below:

- Learning and knowledge rest in diversity of opinions.
- Learning is a process of connecting specialized nodes or information sources.
- Learning may reside in non-human appliances.
- Capacity to know more is more critical than what is currently known.
- Nurturing and maintaining connections are needed to facilitate continual learning.
- Ability to see connections between fields, ideas, and concepts is a core skill.
- Currency (accurate, up-to-date knowledge) is the intent of all connectivist learning activities.
- Decision-making is itself a learning process. Choosing what to learn and the meaning of incoming information is seen through the lens of a shifting reality. While there is a right answer now, it may be wrong tomorrow due to alterations in the information climate affecting the decision. [32] (p.103)

**Table 3.** Lesson Outline: Understanding key characteristics of 'Grounded Theory' as a qualitative methodology v2.

| Lesson Segment | Description | Resources |
|---|---|---|
| Pre-class learning (online, asynchronous) | Tutor assigns relevant methodology readings on Grounded Theory (GT). Additional readings in the form of journal articles are also assigned. Short video clips are assigned to provide a quick introduction to GT. Use of formative assessment, in the form of self-assessment quiz was included. Graduate students who were able to obtain a certain percentage were exempted from going through all of the video resources. Video resources also had interactive questions to make them pause and think about what the video advocated and what were the key concepts that they had to pay attention to and understand. Tutors set up an instant messaging chat group to provide support to the Graduate students for various reasons: (i) technical difficulty (ii) Unable to access the resources (iii) Ease of reach to clarify the tasks they had to do or to clarify doubts. An online sticky note real time collaborative platform was set up. This was an alternate place for Graduate Students to raise questions they had. This online platform served to help build up a class community where they were comfortable sharing with each other, uploaded their additional resources and responded to each other's queries or extended the discussion surrounding the article they had to read. | Pre-class learning (online, asynchronous, use of Learning Management System) Instant messaging chat Online sticky note platform Video resources with interactive quiz |

Blended Learning Approach:
Course level Blending—Graduate students are given the opportunity to immerse with the resources at their own time
Key characteristics of Connectivism:

(i) Diversity—Graduate Students had access to resources of various modalities. They could select resources that they best felt complemented their learning needs before proceeding to others. The various modes of assessment allowed Graduate Students to respond to showcase their learning. Their ability to post questions and responses on the online sticky note platform and the opportunities to ask questions via the instant messaging platform are some examples of how the characteristic of diversity was factored into the design of the cully online learning mode so as to cater to Graduate Students various learning needs.

(ii) Openness—The provision of the various platforms (online sticky notes and instant messaging) allows Graduate students to share their learning and thought processes and to learn from each other. It allows them to openly share their additional resources that they feel might help with their learning as well as their peers learning.

(iii) Connectedness—by providing Graduate students an opportunity to be part of the instant messaging group, allowed them to reach out to their peers to clarify, learn from each other and to reach out to the tutors when they were struggling to grasp some concepts. If they needed help, the platforms were there for the Graduate students to reach out to the knowledgeable others. Library resources and links were made available to them so that they can use the affordance of working online to connect readily to these resource platforms.

(iv) Autonomy—Graduate students had the ability to pace themselves. They could choose how much they wanted to learn at a certain point in time. They also could choose which resources they wanted to begin their learning with. They could choose, after completing the evaluation quizzes as to which materials they needed to access and which were optional for them.

**Table 3.** *Cont.*

| Lesson Segment | Description | Resources |
|---|---|---|
| During class (Synchronous Virtual Class) | Having monitored Graduate students' responses to the quizzes, posts made on the online sticky note and questions and comments raised via instant chat message, the tutor is now able to adopt a targeted approach to clarifying Graduate students doubts and to consolidate the key learning points. Tutor creates breakout rooms and assigns the graduate students to various working groups to critically evaluate an article. This activity will now be complemented by an online productivity platform which allows for real-time collaboration. While Graduate students are in the breakout rooms, they can share screen and work real time collaboratively to critique the article to apply the key concepts they have learnt. Tutor now has the ability to monitor the groups as they work on the templates and guiding questions provided to them. Tutor has the opportunity to encourage the groups who are progressing well via the chat in the productivity tool. Tutor is also able to comment and provide feedback on their responses. Tutor also has to opportunity to join the breakout rooms if the Graduate students reach out and request assistance. Alternatively, tutor can make the decision to join a breakout room to provide support to the group. It is possible to integrate Activity-level blending within this lesson design. This allows the Graduate students with opportunities to refer to resources to engage in self-directed learning. | Online Virtual Meeting platform (to support Virtual Classroom), Chat messaging feature within the platform, ability to show emotions using the features within the platform, Cloud productivity tool for group work |

Blended Learning Approach:
Activity-level Blended Learning—Graduate students have the opportunity to access the online resources, while working in groups, to reference the resources in the event that they have doubts about what the key characteristics of GT as a research design.
Key characteristics of Connectivism:

(i) Diversity—Graduate Students had access to resources of various modality to communicate with their peers and tutor. They could utilize the features within the virtual meeting platform to communicate, they would use the instant messaging chat to reach out to each other or they could simply unmute themselves to share their views.

(ii) Openness—The instant feedback from tutor while they were working on the cloud productivity tool helped them to learn from the tutor and to respond. They could provide feedback to each other instantly by unmuting themselves or via the could productivity tool.

(iii) Connectedness—the breakout room allowed the Graduate students to connect to each other, away from the larger whole class setting. This allowed them to interact within their groups closely and to deepen their learning though their sharing and interaction. By being aware that the tutor was contactable within the virtual platform, through instant messaging or even via the Cloud productivity seemed to have created a more connected learning environment for the students despite the absence of physical interaction.

(iv) Autonomy—Graduate students had the autonomy to either discuss as a group by unmuting themselves or to chat with each other by typing their messages. They had the choice of participating verbally or via chats during the class discussion. The choice of how they wanted to contribute to the group work, allowed the Graduate students options as to how they wanted to contribute to their group work.

| Lesson Segment | Description | Resources |
|---|---|---|
| During class (Synchronous Virtual Class) | Graduate students were requested to return from their breakout room discussion and to return to the main room. At this juncture, they were given opportunities to share their group's responses with others. The others had the option to send in their queries to the group presenting via the virtual meeting chat feature or to unmute themselves and to pose the questions when given the opportunity. In order to minimize online fatigue, tutor also created online form for the others to share their feedback. Tutor collated the feedback and shared it with the respective groups. | Online Virtual Meeting platform (to support Virtual Classroom), Chat messaging feature within the platform, ability to show emotions using the features within the platform, Cloud productivity tool for group work, |

**Table 3.** *Cont.*

| Lesson Segment | Description | Resources |
| --- | --- | --- |
| Blended Learning Approach:<br>Activity-level Blended Learning—Graduate students have the opportunity to access the online resources which other groups had been working on. They had continued access to the resources that were allocated to the groups prior to class.<br>Key characteristics of Connectivism:<br><br>(i) Diversity—While listening to their peers' sharing, the Graduate students have the option of looking at the shared screen or they had the option of looking at the actual files via the cloud productivity tool.<br>(ii) Openness—During the course of the sharing, when opportunities were provided, they could unmute themselves and clarify their understanding or they could further probe into a sharing by a group. If they did not feel like verbally sharing, they could post their viewpoints via the feedback from the tutor had created for this purpose.<br>(iii) Connectedness—Even coming together as a whole class, the Graduate students had opportunities to connect via the various platforms—instant messaging, chat feature within the virtual platform, as well as to comment directly on the presentation materials.<br>(iv) Autonomy—Graduate students had choices they could exercise. They could clarify using direct verbal method or they could use chat features. They could provide feedback verbally or through commenting feature in the cloud productivity or via the form created by the tutor. | | |
| Post-class | Focus is on the application of what the Graduate students have learnt during tutorial, and they have the opportunity to design for a possible area of research applying GT design principles. Graduate students work on their own, at their pace. They can access the materials and resources posted by the tutor for the pre-tutorial activity as well. | Learning Management System, instant messaging, cloud productivity tools and online sticky note platform. |
| Blended Learning Approach:<br>Course Blended Learning—In order to reduce online fatigue, Graduate students now have the opportunity to work on applying the research design to an idea they needed to conceptualize. They could embark on this task on their own time, adopting self-pacing and to contact each other or the tutor if they need guidance.<br>Key characteristics of Connectivism:<br><br>(i) Diversity—Ease of access to various resources available through the Learning Management System. Ease of moving around and self-pacing themselves instead of sitting through a dedicated virtual classroom session.<br>(ii) Openness—Even though the Graduate students were working on their own and self-pacing themselves, the design of the lesson and the availability of resources and channels of communication provided Graduate students with the opportunities to continue to share their learning via the various platforms that have been weaved into the course to support their learning.<br>(iii) Connectedness—While there was no dedicated time set aside for this activity, the availability of the various mediums of communication allowed the Graduate students to be connected to their classmates as well as their tutor.<br>(iv) Autonomy—The Graduate students could exercise choice as to how they wanted to complete the activity. They could complete the task at any time within the deadline set for them. They could get together in groups to discuss and complete their tasks. Due to the online nature of the lesson and the connectivity that was provided, the Graduate students had complete autonomy as to how they wanted to embark on this task. | | |

Based on these principles, connectivism posits that learning occurs when knowledge is actuated by learners connecting to and participating in a learning community. Defined as 'the clustering of similar areas of interest that allows for interaction, sharing, dialoguing and thinking together' [30] (p.3), learners participate in such learning communities and interact among themselves and with others who are more knowledgeable. Such interactions are considered as networks. The key characteristics of such networks, as described by [33], should be given due consideration by educators when designing for learning mediated by technology platforms especially when adopting Blended Learning approach.

(i) Diversity

As described by Downes (2010) [33], the characteristics of 'diversity' requires educational resources to be structured to provide maximum diversity for the learners. Educators should then take into consideration, when designing for online synchronous and asynchronous access that the activities and tasks surrounding the learning of the resources allows for learners to experience creativity, ability to hone into their strengths and to learn using multi-modal resources that help them to learn with confidence, in the best possible

manner for them. Educators, in selecting resources, should focus on multi-modal interface and allow for learners to exhibit evidence of learning through the use of creative and alternative assessment.

(ii)  Openness

Downes [33] describes participants should be able to navigate freely and to be able to access and share free flow of ideas and artifacts within the system. When designing for learning, educators should provide opportunities for learners to share their ideas and be able to share knowledge. While the educator may be the main source of content knowledge, the learners should also be given opportunities to share their resources. Communication should be seamless and need not be only initiated by the educator but could be championed by the learners. This, thus, brings to attention the selection of technology platforms and tools with affordances that will support 'Openness' and for the design of the learning to have tasks and activities that will allow for learners to share freely.

(iii)  Connectedness

Connectedness is seen as being able to access and learn from various communities or known as 'nodes' in connectivism [31]. As educators design for learning, they need to be aware that learners need to be able to learn not only within the community but also be able to access other communities or resources. These could be experts, industry partners, libraries and other resource portals which will help facilitate learning.

(iv)  Autonomy

Autonomy in a connected environment is viewed as the learner having choices or options and having the control to be able to make the choices that best suit their learning [33]. In this regard, educators not only need to ensure that there is a wide array of resources and learning opportunities available to the learner; they must ensure that the learner is not evaluated for making choices that may not seem popular or obvious. Learners need to be empowered to make the choices that best suit their learning needs, especially in a technology-mediated environment where information is readily available at the 'finger-tips' and in various modalities.

The re-designed lesson example in Table 3 will discuss how the authors took into consideration the characteristics as advocated by Connectivism and included learning activities, platforms and opportunities to ensure that learning by graduate students was not in any way compromised with a completely online learning environment.

**5. Key Design Considerations**

The above discussion draws particular attention to the design of learning adopted by educators. Educators need to take into consideration: (i) the profile of their learners; (ii) content being learnt; (iii) pedagogy adopted; (iv) technology tools used; and (v) context where learning is taking place; as well as (vi) assessment. We represent this with a visual representation as depicted in Figure 1.

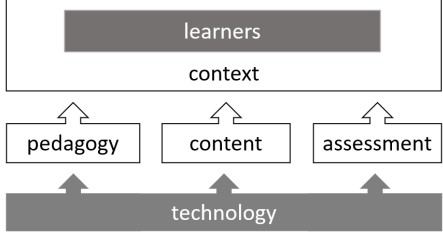

**Figure 1.** Designing for Blended learning: a revised proposition.

The learners remain at the center of all that we do in IHLs. Hence, when we design for learning, knowing their profiles and needs are of utmost importance. The other key consideration is the context. Given the fact that this Blended Learning approach takes

place in situations where physical interaction is not possible, the context then becomes a significant consideration. With no opportunities to meet physically, educators will have to decide on the repertoire of technology tools that will help provide continued support for their learners. Interactions will have to be mediated though the use of technology tools, thus impacting the decisions educators have to make in relation to the type of pedagogy they are adopting and their choice of modalities, e.g., asynchronous or synchronous. Adopting the relevant pedagogy supported by appropriate technology tools also becomes a key area of focus. Educators need to take into consideration technological affordances that support the pedagogical design of the lesson, the availability of tools as well as connectivity available to the learners. Online modes of lesson delivery also prevent instructors from effectively observing learners' emotions and body language. While assessment still remains an indispensable aspect of teaching and learning, educators will have to make efforts to include creative forms of assessment that will help learners reflect on their own learning and provide information/data to educators about their progress. In this adapted Blended Learning milieu, it is important to note that assessment is influenced by social cognition, multimodal texts and ubiquitous environment [34] These creative alternatives can go to some extent in reducing the fatigue caused by formal/summative assessment as educators harness the affordance to creatively design for monitoring of learning. It is of paramount importance to note that all of the design considerations have to be dynamically designed with the support of technology tools so as to ensure educators are able to engage students in the learning process.

In summary, Figure 2 below attempts to explicate the contributions made by this concept paper to extend the concept of blended learning. Captured are the adaptations made to the four traditional dimensions in blended learning, i.e., space, time, fidelity, and humanness [20]. We would like to draw attention to Rich* under the virtual column. While fidelity in a physical environment in a blended approach provided for educators to design for a rich learning experience, virtual learning environments used to fall short in this area. Given the rapid advancement in technology and Web 2.0, educators have the oppor-tunity to design for learning virtually with high fidelity, including simulated learning and multi-modal resources. The changes made were necessary in order to respond to the challenges brought about in recent times. The fluid nature of our teaching and learning environments, alongside other key considerations such as learners' online fatigue, shorter attention spans, lack of interactivity, the need for human touch, demanded that we find new ways to approach online/virtual lessons [23].

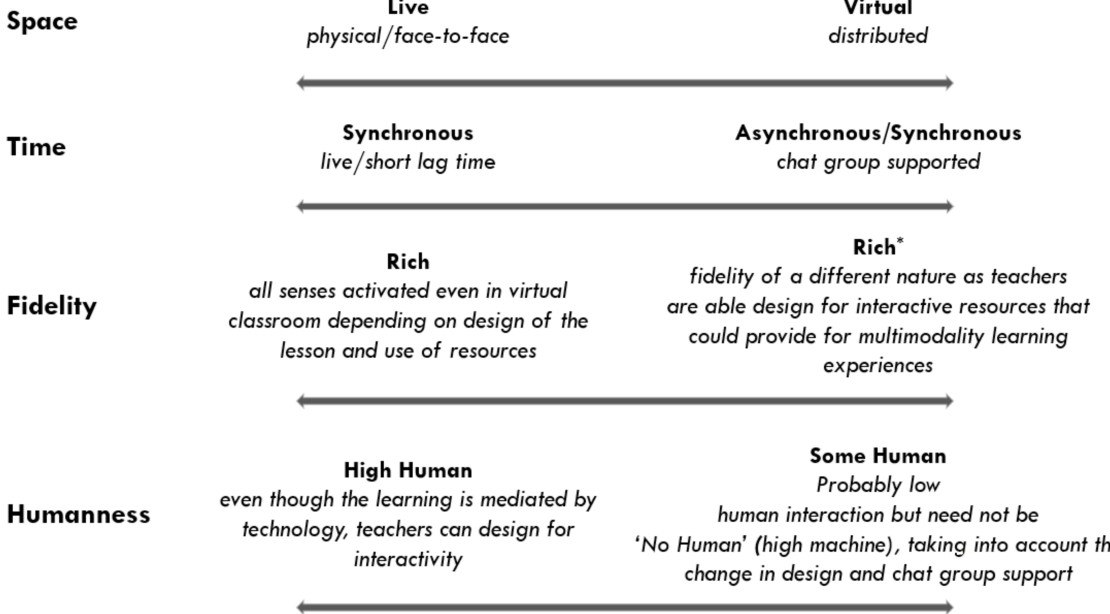

**Figure 2.** Revised dimensions of interaction in blended learning (adapted from [23]).

## 6. Conclusions

This conceptual paper started off arguing for a reimagined approach to blended learning in order to maintain continued education for our learners due to the various challenges brought about by the COVID-19 pandemic. It has shown how it is possible to adopt sound pedagogical design principles adapted from the blended learning approach and connectivism to ensure that the business of teaching, learning and assessment can still continue. As explained in the introduction, the intention was not to propose a new theory but rather to connect existing ones in order to shed light on the challenging circumstances that the pandemic has brought about to our institution and other IHLs across the globe. This is performed in the spirit that fellow educators can draw from our experiences and then broaden their scope of thinking when using technology to mediate learning.

What would be useful to support the ideas introduced in this paper would be empirical data about this reimagined approach to blended learning and its impact on learners. The following are some ideas for research: One, more detailed and fine-grained analyses of a wider sample of lessons designs. Two, a study examining instructors' and learners' perceptions of how revised ways of content delivery have impacted their teaching, learning and assessment. Three, a comparative study of how this blended learning approach might differ from subject to subject—starting with the assumption that a one-size-fits-all approach might not to the best way forward.

The COVID-19 pandemic may have abated in many parts of the world but there are still many education jurisdictions that will continue to face challenges—conflicts, disasters, new and old epidemics, etc. In such situations where physical face-to-face interactions are not possible or when physical learning spaces cannot be made available, this proposed approach can become a viable alternative to ensure learning continues. What this approach requires is that educators make the shift in their mindsets about where and how teaching, learning and assessment can take place. The connected environment that technology affords teachers and learners is a powerful one if exploited in the right spirit with the right motives.

**Author Contributions:** Data curation, S.D.; Formal analysis, A.C. All authors have read and agreed to the published version of the manuscript.

**Funding:** This research received no external funding.

**Institutional Review Board Statement:** Not applicable.

**Informed Consent Statement:** Not applicable.

**Data Availability Statement:** Not applicable.

**Conflicts of Interest:** The authors declare no conflict of interest.

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
