# Peer review of "Blended Learning Reimagined: Teaching and Learning in Challenging Contexts"

_education, doi:10.3390/educsci12100648_

Round 1
Reviewer 1 Report
Thank you for the opportunity to review this manuscript which I found to be generally well-written and interesting. Below, I present a number of suggestion, which aim to strengthen the article further.
1. For for the benefit of the reader, you need to include a section (or a paragraph in the introduction), explaining your method. It’s a conceptual article of course. However, how did you revie the literature? For example, did you use a Qualitative Content analysis, or something similar?
2. On the last paragraph of the Introduction, at the end, perhaps you could add another sentence, explaining the structure of the paper. This would beneficial to the reader, who will be able to formulate a clearer understanding in terms of organization and navigation.
3. Literature Review, Very good use of updated sources. For some best practices/ teaching practices blended and online/ and quality learning, I encourage you to utilise the following sources:
Salas-Valdivia, L. and Gutierrez-Aguilar, O. (2021) "Key factors for the success of connectivism in the elearning modality in the context of Covid-19," 2021 XVI Latin American Conference on Learning Technologies (LACLO), 2021, pp. 368-371, doi: 10.1109/LACLO54177.2021.00044.
Doukanari, E.; Ktoridou, D.; Efthymiou, L.; Epaminonda, E. (2021) The Quest for Sustainable Teaching Praxis: Opportunities and Challenges of Multidisciplinary and Multicultural Teamwork. Sustainability, 13, 7210. https://doi.org/10.3390/su13137210
Efthymiou, L. and Zarifis, A. (2021) Modeling students' voice for enhanced quality in online management education. The International Journal of Management Education, 19(2): 1-16. https://doi.org/10.1016/j.ijme.2021.100464.
4. Conclusion: in the conclusion, perhaps you can a) discuss in more depth the main points of the paper, and b) make a stronger case for the original contributions the paper makes. In other words, why is this study needed now and how does it advance our understanding of relevant theoretical or empirical matters? Also, c) please include a sentence on future research, along with d) the limitations of the paper.
Very good work overall. I look forward to receiving a revised version of the paper.
Cordially,
Reviewer 2 Report
The research presented in this study is about reimagining Blended Learning and is conceptualized by the authors as a ‘concept paper’ (line 12 & line 253). However, the paper presents the redesign of a lesson in the context of a PhD program, switching form a blended approach (pre-Covid), to a completely online course (during the COVID pandemic).
Comments and suggestions
Minor issues: there are some typos, e.g., …
Line 45.- from line 45 to the end of the paper, the text is not properly aligned (in the left side).
Line 54.- ‘deaths [12] Other than’. Is a full stop required (deaths [12]. Other than)?
Line 93.- from line 93 to the end of the paper, the text is not properly justified in the right side, besides continuing not being aligned in the left side.
Line 101.- between line 101 and line 102, too much space.
Line 125.- a final full stop is required in the line.
Line 125 to 132.- too much space between lines.
Line 155 to 158.- Lacks a question mark at the end of the sentence? Lacks ‘:’ between ‘to (i)’? Lacks ‘,’ or ‘;’ before (i), (ii), and (iii)?
Table 2 (line 224) .- ‘to resources of various modality’; in plural, to resources of various modalities.
Line 228 to 231.- Lacks ‘:’ between ‘to (i)’? Lacks ‘,’ or ‘;’ before (i), (ii), (iii), (iv) and (iii)? There are two ‘and’: ‘pedagogy adopted and (iv) technology tools used and (v) context where learning is taking place’
Line 277. reference number 8: there are 2 full stops at the end of the reference.
Line 281.- reference number 10: the font should be changed (name of the authors).
Delete lines that are not required, e.g., 51, 83, 146, 147, 161, 227, 260, …
Some considerations …
References to the different Master Plans for ICT (lines 63, 65, 68, 69) should be included in the article.
Reference to the Educational Technology Plan should be included (line 73).
Line 205.- ‘Connectedness’. Are the ideas of the author, or are ideas of other authors that should be properly referenced? Downes…?
Line 211.- ‘Autonomy’. Are the ideas of the author, or are ideas of other authors that should be properly referenced? Downes…?
Key Design Considerations & Conclusions
In the article the authors explain the strategy followed in Singapore.
In fact, the lesson that is presented was taught previously in a blended format -activity-blended learning, in words of Graham (2006)-, as written in line 111; and once the COVID pandemic forced to change the format, an online format was implemented, again a format that matches to blended learning according to your writing, and that nowadays is referenced in research works as virtual. Then, what is presented as an experience, is a switch from blended to virtual (or to a subtype of blended, according to Graham (2006).
This conceptual work is based, in terms of background, on blended learning -following Graham (2006)- and connectivism.
Line 231 to 234.- It is said that the content of the paragraph (line 228 to line 230) is represented in Figure 1. However, in the previous paragraph to Figure 1 there is no mention to ‘assessment’, despite it appears as one of the key elements included in Figure 1. In fact, assessment (one of the most challenging elements when thinking about the virtual format) is just mentioned in line 245 to line 247, citing that ‘it is still indispensable’.
The main conclusion that is written is ‘this approach become a viable alternative to ensure there is continued learning’ (line 255), besides stating that instructors have to ‘make the shift in their mindsets about where and how teaching, learning and assessment can take place’ (line 256 to 257).
Which one are the novel insights that can be extracted from this concept paper?
References
Some additional and recent references about virtual course could be included in this article to enrich this written contribution.
Some DOI are missed, e.g.
8. Rashid, R., & Yadav, S.S., (2020). Impact of Covid-19 on Higher Education and Research. Indian Journal of Human Development, 277 14(2), 340-343. DOI:10.1177/0973703020946700
27. Bell, F. (2011). Connectivism: Its Place in Theory-Informed Research and Innovation in Technology-Enabled Learning. International Review of Research in Open and Distance Learning 12(3) DOI:10.19173/irrodl.v12i3.902
--
Round 2
Reviewer 2 Report
Line 57.- design., The